# A Review on Material Selection Benchmarking in GeTe-Based RF Phase-Change Switches for Each Layer

**DOI:** 10.3390/mi15030380

**Published:** 2024-03-13

**Authors:** Sheng Qu, Libin Gao, Jiamei Wang, Hongwei Chen, Jihua Zhang

**Affiliations:** 1School of Integrated Circuit Science and Engineering, University of Electronic Science and Technology of China, Chengdu 610054, China; qushengqs@126.com (S.Q.); wangjiameiaaa@163.com (J.W.); hwchen@uestc.edu.cn (H.C.); 2State Key Laboratory of Electronic Thin Films and Integrated Devices, University of Electronic Science and Technology of China, Chengdu 610054, China

**Keywords:** chalcogenides, RF phase-change switches, GeTe, RF performance, thin films

## Abstract

The global demand for radio frequency (RF) modules and components has grown exponentially in recent decades. RF switches are the essential unit in RF front-end and reconfigurable systems leading to the rapid development of novel and advanced switch technology. Germanium telluride (GeTe), as one of the Chalcogenide phase-change materials, has been applied as an RF switch due to its low insertion loss, high isolation, fast switching speed, and low power consumption in recent years. In this review, an in-depth exploration of GeTe film characterization is presented, followed by a comparison of the device structure of directly heated and indirectly heated RF phase-change switches (RFPCSs). Focusing on the prototypical structure of indirectly heated RFPCSs as the reference, the intrinsic properties of each material layer and the rationale behind the material selection is analyzed. Furthermore, the design size of each material layer of the device and its subsequent RF performance are summarized. Finally, we cast our gaze toward the promising future prospects of RFPCS technology.

## 1. Introduction

In contemporary wireless communication systems such as satellites, radar, mobile communication, and electronic countermeasures, a radio frequency (RF) module serves as an integral front-end component. The rapid evolution of millimeter-wave (MMW) technology has spurred dynamic shifts in wireless network requirements, demanding an increased array of devices for smooth integration into wireless networks. For achieving faster data processing rates, the conventional approach involves assembling various RF front-ends tailored for different devices. However, this method tends to demand more space and incurs higher costs, counter to the development ethos of miniaturization and integration. To accommodate diverse frequencies, communication standards, and functional devices, there arises a heightened requirement for reconfigurable modules or systems. Within these reconfigurable systems, RF switches assume a pivotal role as core devices. Considering factors such as integration compatibility, process complexity, and device size, the commonly employed types of RF switches typically include mechanical switches, semiconductor switches, and Micro-Electro-Mechanical Systems (MEMSs) [1,2,3,4,5,6,7,8,9,10,11].

Mechanical switches show exceptional RF performance within the DC-10 GHz range. Nonetheless, these switches exhibit remarkably slow switching speeds, typically ranging from 2 to 50 ms [1,2,3]. Furthermore, they tend to be bulky in size. On the other hand, semiconductor switches are better suited for high-speed switch applications and offer ease of integration into semiconductor substrates like silicon and gallium arsenide (GaAs). Although semiconductor switches like Positive–Intrinsic–Negative (PIN) diodes [4,5,6] and Field Effect Transistor (FET) switches [7,8] are known for their compact size and cost-effective integration into other front-end RF modules, they are also susceptible to distortion, leakage, voltage breakdown, and high non-linearity. Compared with mechanical switches and semiconductor switches, MEMSs have high isolation and good linearity. However, MEMS technology involves a complex process, high power consumption, and often results in lower yields [9,10,11]. A recent innovation, the GeTe-based phase-change switch (PCS), has emerged and exhibits promising potential as an RF application candidate. PCSs boast a host of advantages, including superior RF performance, high linearity, enhanced integration capabilities, non-volatile characteristics, and rapid switching speeds. These highly competitive properties position PCSs as an ideal choice for miniaturized and intricate RF components, particularly in millimeter-wave applications. In this paper, the focus is primarily on phase-change switches (PCSs) driven by electrical pulses for switching transitions, while another category of PCSs utilizing optical pulses, specifically laser, will not be extensively discussed. Optical-activated PCSs boast a remarkable feature of ultra-fast switching within 200 ps, coupled with a relatively simple structure compared to electrical-activated PCSs [12,13,14,15,16,17]. However, future integration poses a challenge due to the requirement of a laser source. On the other hand, electrical-activated PCSs, categorized based on heating methods, include direct heating [18,19,20,21,22] and indirect heating [23,24,25,26,27,28]. These two types of switches will be discussed in more detail in later chapters.

While there exists a burgeoning report on GeTe-based PCSs, the detailed characteristics of the selected material in each layer remain insufficiently explored. This review aims to consolidate recent edges in phase-change RF switches and comprehensively analyze the structural composition of these switching devices concerning materials. The goal is to significantly augment the future RF performance of PCSs by refining design parameters and optimizing material selection. The subsequent content delves into an in-depth analysis of the materials used in each layer of a PCS, followed by an outlook on future prospects for this technology.

## 2. Selection of Phase-Change Material

### 2.1. The Phase Change of Chalcogenides

Phase-change materials (PCMs) exhibit a distinctive characteristic whereby they can undergo reversible transformations between the amorphous and crystalline states following heat treatment with precise electrical pulses, as depicted in Figure 1a [29,30,31]. The amorphous configuration signifies a disorganized arrangement of atoms, referred to as short-range ordering, whereas the crystalline state reflects an organized atomic arrangement, known as long-range ordering. The disordered amorphous state features a reduced mean free path for electrons, impeded by scattering, consequently resulting in higher electrical resistance relative to the crystalline state.

When subjected to intense and short-duration pulses, the material’s temperature rises rapidly above its melting point (T_m_) and changes into a liquid state. Subsequent rapid quenching prompts the atoms to solidify with a disordered arrangement, thus completing the transition from the crystalline to the amorphous state. In contrast, the transition from amorphous to crystalline necessitates maintaining the temperature of PCM below the melting point and keeping it above the crystallization temperature (T_c_) for a certain duration, allowing atoms to reassemble into an organized arrangement, as illustrated in Figure 1a [32,33].

**Figure 1 micromachines-15-00380-f001:**
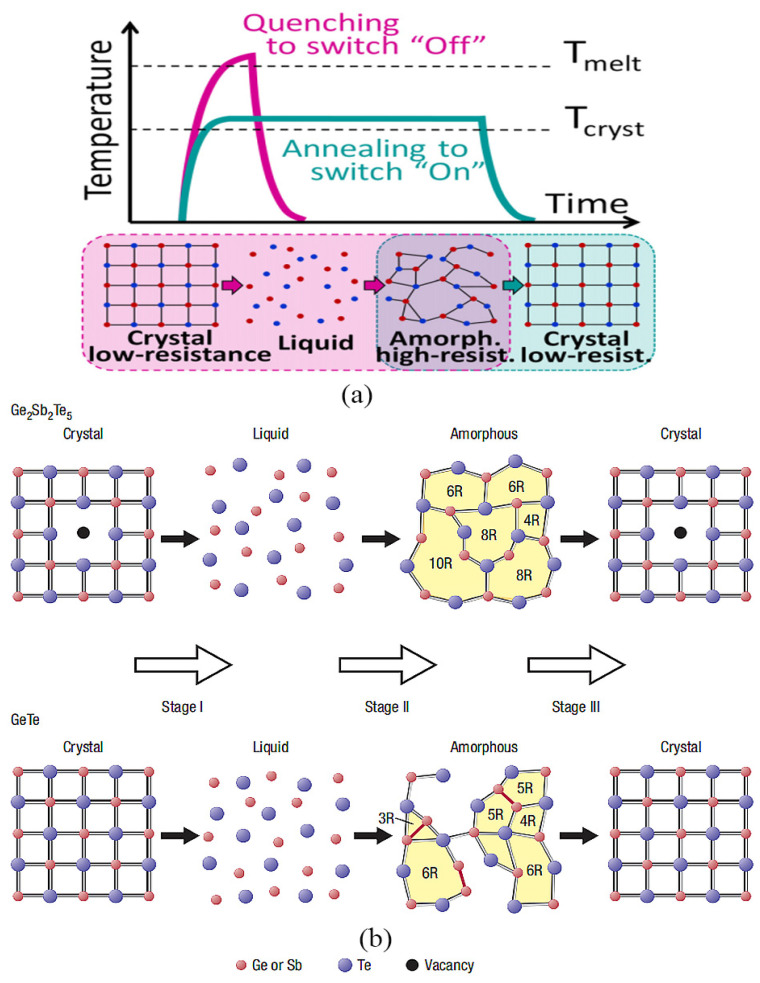
(**a**) Reversible switching of PCM under thermal pulses (adapted from [31]). (**b**) Schematic presentation of the possible ring structure transformation in the phase changes of crystal–liquid–amorphous and amorphous–crystal in Ge_2_Sb_2_Te_5_ and GeTe (adapted from [33]).

Typically, the amorphous state is a highly disordered arrangement of atoms, but the amorphous state of chalcogenide compounds contains some interesting rules of arrangement. Kohara et al. conclude that amorphous Ge_2_Sb_2_Te_5_ is characterized by even-folded ring structures, whereas amorphous GeTe has both even- and odd-numbered rings [33]. A diverse array of ring structures exists in GeTe, encompassing both even- and odd-numbered formations. As illustrated in Figure 1b, the presence of odd-numbered rings predominantly signifies the existence of homopolar bonds like Ge-Ge bonds, while the prevalence of even-numbered rings indicates a pronounced inclination towards heteropolar bonds, thereby suggesting the absence of Ge-Ge bonds in Ge_2_Sb_2_Te_5_. The intricate calculations performed further unveil microscopic remnants of the crystal structure within the amorphous state, such as the persistence of even-numbered rings and bond angles approaching 90 degrees. It is noteworthy that although the total pair correlation functions between the amorphous and crystalline states significantly differ, the amorphous structure retains several distinctive features reminiscent of its crystalline counterpart.

The potential materials mainly used in RF switches are Ge-Sb-Te (Ge_2_Sb_2_Te_5_), germanium tellurium (GeTe), antimony telluride (SbTe), and vanadium oxide (VO_2_). Currently, a great deal of research work endeavors concerning these materials has predominantly centered around optical storage as well as electrical storage in phase-change memories [34,35,36,37,38], and some of them have already been realized for a wide range of commercial applications, like CD and DVD. In recent years, a growing number of scholars have extended the utilization of these materials towards achieving the functionality of RF switches [12,13,14,15,16,17,18,19,20,21,22,39,40,41,42,43,44].

Moon et al. reported the first SbTe phase-change material RF switch with a cutoff frequency (F_co_) of 4.1 THz [39]. Subsequently, Yalon et al. proposed an indirect heating phase-change RF switch based on Sb_7_Te_3_, with F_co_ of 5 THz, and power and energy density of ~8 mW/μm^2^ and ~2.4 nJ/μm^2^, respectively [40]. Mennai et al. presented the first Ge_2_Sb_2_Te_5_ phase-change material RF switch with the figure of merit (FOM) of about 450 fs [41]. Pan et al. designed the VO_2_-based single-pole-single-throw(SPST) switch with the insertion loss < 3 dB at 10 GHz. In addition, the isolation of SPST switch improved to better than 30 dB [42]. Given that GeTe is the most used phase-change material on RF switches, a specific discussion will follow in a later section. Detailed descriptions of the performance about specific switches will be provided in subsequent chapters. Table 1 illustrates the performance outlined in pertinent studies of RF switches. Considering the advantages of phase-change temperature, resistivity, and non-volatility, GeTe thin films dominate RF PCSs, according to the reported literature.

### 2.2. The Structural Analysis of GeTe

GeTe is categorized as a degenerately doped p-type semiconductor based on its electronic properties. It exhibits two stable states at room temperature, which can be switched repeatedly. As far as the structure of crystalline GeTe is concerned, it was initially hypothesized to correspond to an ideal rock salt structure, commonly referred to as rhombohedral GeTe (R-GeTe). However, it was found that R-GeTe is a distorted face-centered cubic (FCC) as rhombohedral structure, as shown in Figure 2a. GeTe begins to undergo an ordered–disordered transition at a temperature range of 473 K to 673 K [45,46]. When it undergoes homogeneous melting, R-GeTe transitions to cubic GeTe (C-GeTe), which is formed at 693 K and is observed up to 973 K [47]. In the rhombohedral configuration, the Ge atom resides off-center within the octahedron formed by six Te atoms with 3 longer bonds and 3 shorter bonds as red sticks and green sticks as Figure 1a shows, while in the rock salt as well as cubic arrangement, it occupies the central position of the octahedron. In contrast, all Ge-Te bonds exhibit uniform length in the cubic structure, as 6 blue sticks showcased in Figure 2b [48].

Figure 2c shows the Transmission Electron Microscope (TEM) image of GeTe films deposited at 573 K and annealed for 24 h. The corresponding fast Fourier transform (FFT) patterns (insets in Figure 2c) of the GeTe film indicate a rhombohedral crystal structure (space group R3m). FFT exhibits lattice fingers spaced 3.29, 3.03, 2.10, 1.76, 1.44, 1.88 Å and 1.18 Å from the (021), (202), (024), (006), (404), (205) and (009) planes of the R-GeTe crystal structure (space group R3m), respectively [49]. Figure 2d depicts the high-resolution TEM image of GeTe film annealed in a vacuum furnace at 673 K. In the TEM of crystallite, two sets of planes are observed. Besides the (111) lattice planes with a plane spacing of 3.5 Å, the other exhibits a series of (200) planes with a lattice spacing of 3.0 Å, intersecting with the (111) planes at an angle of 55°. The corresponding fast Fourier transform (FFT) patterns (insets in Figure 2d) of the GeTe film reveal a face-centered cubic pattern [50], confirming the rock salt structure of the crystallite. Across all crystallites, the (111) planes align parallel to the film surface, suggesting that the preferential orientation of the crystallites within the C-GeTe film lies in the (111) direction along the film thickness. This aligns with previous reports [46].

Figure 2e offers a comparison between the calculated band structures of R-GeTe and C-GeTe. Following the phase transition, the calculated bandgap of GeTe decreases from 0.56 eV (indirect type) in R-GeTe to 0.37 eV (direct type) in C-GeTe [51]. The bandgap in the amorphous state has been reported to be approximately 0.8 eV. This relatively wide bandgap restricts the mobility of charge carriers and thus limits the conductivity of the material. However, upon transitioning to the crystalline state, the bandgap narrows significantly. This narrowing of the bandgap facilitates the movement of charge carriers, leading to a substantial enhancement in the conductivity of the GeTe crystal. Consequently, the narrow bandgap in the crystalline state plays a pivotal role in enabling efficient charge transport within the material, thereby greatly improving its overall conductivity, especially in C-GeTe. The calculated Fermi surfaces in Figure 2f reveal similar crystal structures for R-GeTe and C-GeTe, with nearly identical first Brillouin zones. Additionally, R-GeTe encompasses six full valleys of VBΣ, while C-GeTe comprises twelve full valleys of VBΣ plus eight half valleys of VBL. Consequently, the rhombohedral-to-cubic phase transition significantly augments the band degeneracy in GeTe. Therefore, C-GeTe exhibits superior electronic-transport properties compared to R-GeTe [52,53].

X-ray absorption fine structure (EXAFS) measurements reveal that throughout the amorphization process, the average coordination number of Ge ions decreases from six to four, as demonstrated in studies [54,55]. The motif of amorphous GeTe is rigorously supported by Figure 3a,b [56,57]. Kolobov et al. postulated that Ge resides in a hybrid structure combining aspects of a tetrahedral configuration and a defective octahedral configuration within amorphous GeTe [46]. This structural arrangement undergoes a remarkable transformation throughout the crystallization process, transitioning from a tetrahedral form to a defect-free octahedral form. It is important to note that amorphous GeTe does not exhibit complete disorder. Yannopoulos et al. have studied the temperature-dependent Raman spectra of GeTe [57]. In Figure 3c, the Raman spectra of crystalline GeTe exhibit just two distinctive peaks, aligning with the amorphous B and C peaks (offset by ~45 cm^−1^). Remarkably, the observed proportional relationship between frequency and energy, E (d^cr^) < E (d^a^), aligns with the proposed formula E (where d^cr^ and d^a^) = 0.635E (d^a^), where d^cr^ and d^a^ represent crystalline and amorphous bond lengths and E represents energy [58]. The data depicted in Figure 3d, based on calculations, illustrate an increase in the ratio between the integrated intensity of the B_1_ and B_2_ bands, as well as the corresponding ratio of the C_1_ and C_2_ bands when subjecting the material to heat [59]. This observation infers a gradual shift in the composition of the tetrahedral species forming the glassy structure as the amorphous film is heated. This shift leads to the formation of Te-rich tetrahedra at the expense of Ge-rich tetrahedra, thus implying that the atomic arrangement within the primary coordination unit changes as a result of the temperature increase, even below T_c_. A quantitative analysis of the vibration modes was carried out using a Gaussian line fitting method, revealing the presence of five distinct vibration modes. Therefore, the possible tetrahedral combination structure of GeTe_4−n_Ge_n_ (n = 0, 1, 2, 3, 4) was proposed [57]. This remarkable outcome suggests significant structural re-configurations occurring within the amorphous solid structure, even at temperatures lower than the crystallization temperature. The amorphous structure is delineated by a composition of five tetrahedral combinations: 6.25% GeTe_4_, 25% GeTe_3_Ge, 37.5% GeTe_2_Ge, 2.25% GeTeGe_3_, and 6.25% GeGe_4_. The relative distribution of these five structures is illustrated in Figure 3e.

Ab initio molecular dynamics (AIMD) modeling of amorphous GeTe obtained by quenching in the melt revealed a substantial presence of tetrahedrally coordinated germanium atoms. These tetrahedral structures seldom comprise four Te ligands. However, they consistently include at least one homopolar Ge-Ge bond, as sketched in the upper part of Figure 3f [60]. Within tetrahedra, the bond-weighted distribution function (BWDF) of homopolar Ge-Ge contacts (bottom left part of Figure 3f) exhibits a more significant bonding peak at short distances of approximately 2.6, demonstrating that Ge-Ge bonds within tetrahedral formations appear chemically strong, while in octahedral environments, the bottom left part of Figure 3f indicates a lack of significant Ge-Ge bonding as the positive BWDF region is counteracted by an anti-bonding one at longer distances. Regarding heteropolar bonds, the Ge-Te contact nature appears relatively consistent regardless of the local structure, displaying similar BWDF curves in bottom right part of Figure 3f. This suggests that homopolar bonds bolster tetrahedral configurations in amorphous GeTe but do not notably fortify octahedral arrangements. Left-hand p-COOP (partial Crystal Orbital Population) curves in Figure 3g reveal pronounced instabilities in the tetrahedral GeTe_4_ motifs, notably featuring a major anti-bonding peak at the εF position. This aligns with the infrequency of pure GeTe_4_ tetrahedra observed Figure 3f. Conversely, the presence of one or two homopolar bonds (middle and right-hand panels in Figure 3g) significantly diminishes these antibonding interactions within the tetrahedra. Judging solely from the shape of the p-COOP curves, homopolar contacts seem more conducive to the tetrahedral motif [61].

Recently, it has been argued that the observed “ideal” rock salt structure is the result of the averaging effect, not the result of the local atomic environment. Even at high temperatures, the length of the Ge-Te bond will not be affected [56]. The source of this obvious difference lies in different measurement techniques: X-ray and neutron diffraction produce the overall structure, while EXAFS detects the local surroundings of the atom. As far as the current study is concerned, the precise nature of the phase transition remains a topic of ongoing discussion [62].

### 2.3. Characteristic of GeTe Films

The stoichiometric balance between Ge and Te in GeTe alloy compounds significantly impacts resistivity. GeTe films with diverse composition ratios exhibit considerable variations in crystalline and amorphous resistivity [63]. Typically, the ratio of amorphous resistivity to crystalline resistivity spans three to six orders of magnitude. In the conducted temperature-dependent resistivity experiments on 100 nm thick Ge_x_Te_100−x_ films prepared by magnetron sputtering with different GeTe compositions, depicted in Figure 4a,b, it is notable that the Ge_36_Te_64_ film displays the most pronounced resistivity shift between its amorphous and crystalline states, alongside minimal programming requirements [64,65]. The Ge_44_Te_56_ film demonstrates the lowest resistivity, with crystallization occurring at approximately 190 °C. In Figure 4c,d, X-ray diffraction (XRD) analyses of the five compositions are depicted. For Te-rich samples like Ge_36_Te_64_ and Ge_44_Te_56_, as shown in Figure 4c, the crystalline rhombohedral GeTe and hexagonal Te evidence appears at 250 °C. GeTe and Te crystallization happens simultaneously at the same temperature. In proximity to the 50:50 stoichiometric composition, the Ge_53_Te_47_ sample displays only characteristic rhombohedral GeTe peaks. Conversely, for Ge-rich samples, as shown in Figure 4d, the segregation of cubic Ge within a crystalline GeTe matrix is confirmed at temperatures surpassing 325 °C. This is illustrated by the emergence of (111) and (220) peaks characteristic of cubic-Ge in Ge_61_Te_39_ and Ge_69_Te_31_. Raoux et al. have demonstrated the sequential crystallization of GeTe and Ge, with the latter occurring at a higher temperature [66]. GeTe films exhibit diverse states due to control over deposition factors such as temperature, applied electric field on the substrate, seed layer structure, and deposition techniques (i.e., evaporation, sputtering, and epitaxial growth) [63]. GeTe films deposited below 130 °C typically exhibit an amorphous state. Deposited GeTe films adopt a polycrystalline state within the temperature range of 130 °C to 250 °C. If grown on a NaCl or mica seed layer at nearly 250 °C, GeTe forms a crystalline state with a rhombohedral structure. Above 250 °C, a crystalline GeTe film with a NaCl (or FCC) structure is observed [67,68,69].

By utilizing a 50:50 target for GeTe and adjusting the sputtering power and pressure, films can be produced with varying resistivity and crystallization temperature. GeTe films deposited at room temperature are initially amorphous. The crystalline GeTe film can be obtained by sputtering at elevated temperature or by annealing the as-deposited film. Sputtering power, deposition pressure, and Argon flow contribute to the resistance of GeTe, as shown in Figure 5a,b [70,71]. It is evident that higher sputtering power and lower pressure lead to lower crystalline resistance in most cases. In addition, the crystalline temperature changed as well with a different sputtering parameter. Figure 5c shows the scanning electronic image (SEM) of as-deposited and crystallized GeTe films [71]. The average grain size before crystallization is about 20 nm, but after crystallization the grain size increases by an order of magnitude (100–700 nm). The authors assumed that formation of voids is common in crystalline films, which may be due to densification within the film grains and the coalescence of micrometer- or nanometer-sized voids between grains. However, they have demonstrated that these voids do not have a deleterious effect on device performance when applied to PCSs with thees films. However, they have demonstrated that these voids do not adversely affect device performance when applied to PCSs with the film. Figure 5d illustrates the changes in the crystal state of GeTe during the annealing process [70]. As the temperature increases, the surface of GeTe will be the first to produce oxides of Ge and Te, followed by the formation of a thin oxide layer on the surface, and only below the oxide layer is a thin film of crystalline GeTe. However, if there is a passivation layer on the surface of GeTe, it is easy to avoid oxidation during annealing in the process of fabrication.

## 3. GeTe-Based Phase-Change Switches (PCSs) for RF Application

In recent years, GeTe has been used as PCSs in RF applications, showing great potential due to its advantages, as seen in Table 1. According to the method of heating, phase-change switches are mainly divided into direct heating structures and indirect heating structures. The structure and performance of the two switches with different heating methods will be compared in next section.

### 3.1. Comparison with Direct and Indirect Heating Structure Phase-Change Switches

This paper is the first to systematically analyze the stack structure of indirectly and directly heated phase-change RF switches, and analyze the roles of each functional layer other than the phase-change material, which will provide a reference for the optimization of device performance and future designs.

Figure 6a,b show the schematic diagram of the two types of stack structure for phase-change switches. In this review, GeTe is the constant phase-change material due to advantages for RF application. Both switches start with substrate and dielectric layer (bottom). In order to accommodate back-end-of-line (BEOL) processes for further integration, substrates such as HR-Si, Sapphire, SiC, GaAs, and GaN are optional. SiO_2_, SiN_x_, and AlN film are also commonly used as dielectric layer for good insulation. A heater is used in both structures for raising the temperature of the phase-change layer. The phase-change transition results from the temperature rapidly rising and quenching or being kept continuous. The distinction between the two switches lies in the current flow mechanism: in a directly heated switch, the current flows through GeTe (the PCM) to heat itself, whereas in an indirectly heated switch, the current passes through a microheater that generates heat, which is then conducted to the GeTe [18,23]. Materials used for the heater are metals such as TiW, TiN, NiCrSi, NiCr, W, Mo, etc. A stacking structure is used in an indirectly heated switch, as depicted in Figure 6b. The metal electrodes for the RF path are different combinations of Au or Cu and adhesion metals such as Ti, TiW, Cr, W, etc. The pros and cons of common materials used for each layer based on works reported so far will be analyzed.

### 3.2. The Direct Heating Structure of Phase-Change Switches

Chua et al. first proposed a reconfigurable switch based on germanium tellurium in 2010 [70]. The GeTe reconfigurable phase-change switch has a low on-resistance of 180 Ω and a large dynamic range of 7.2 × 10^3^. The GeTe switch has a diameter of 1 um and a thickness of 100 nm. The joule heat induced by the pulse current can convert the on-state to the off-state. The transition from off-state to on-state is achieved by Poole–Frenkel effect or the filament conductivity model [73,74,75]. The filament conductivity model used in the ovonic switching method explains the variation in R_ON_, and a possible reason for partial re-amorphization and partial crystallization of the switch for the reduction in the dynamic range was proposed. The authors believe that this GeTe switch could be sufficient for many RF applications.

In 2013, Shim et al. [22] conducted a study on the RF characteristics of germanium telluride in phase-change switches. The research focused on critical RF performance of the switches, such as the third-order intercept point (IIP3) and the 1 dB compression point (P1dB), analyzing their behavior. The initial utilization of GeTe as a switch in the RF domain was realized, resulting in some observed RF performances. The extracted IIP3 in the crystalline state stood at 37 dBm, while in the amorphous state, it was approximately 33 dBm, with a frequency offset of 50 kHz at 3.9 GHz. Notably, the switch’s linearity exhibited degradation at lower frequencies.

In 2014, Wang and Rais-Zadeh [23,24] introduced a directly heated four-terminal switch structure featuring dual RF ports (input and output) and a dedicated path for the heater. This switch exhibited an on-state insertion loss of less than 0.5 dB and off-state isolation exceeding 18 dB at frequencies up to 20 GHz, highlighting a cutoff frequency surpassing 4 THz. With a measured P1dB considerably greater than 20 dBm and an IIP3 exceeding 30 dBm, this architecture displayed impressive performance characteristics.

In 2019, Léon et al. [25,26] compared two configurations utilizing germanium telluride (GeTe) phase-change-material-based direct heating switches. Employing deep UV lithography stepper technology enabled the definition of consistent patterns with widths as narrow as 250 nm. For the first time, design guidelines were established linking the material geometry to the maximum allowable current through the switch before failure. The study concluded that direct heating offers an efficient solution for GeTe amorphization, effectively preventing heater aging. Notably, in both configurations, a thick and short GeTe switch geometry was found to be advantageous. The shunt configuration achieved 31 dBm in the on-state and over 35 dBm in the off-state, while the series configuration achieved 27 dBm in the on-state and 32 dBm in the off-state. Moreover, the cutoff frequency for the shunt and series configurations reached 11 THz and 21 THz, respectively, at frequencies up to 40 GHz.

Figure 7a–c show three main models of direct heating structures: two-port and four-port sandwich structures and a four-port planar structure. Figure 7a is considered as a sandwich structure with a GeTe layer located in the middle of the heating electrodes. Figure 7b,c are schematic images of these sandwich structure of fabricated direct heating PCSs. Figure 7c shows the planar connection structure of direct heating and the two configurations of shunt and series switches were applied in this configuration [19]. This structure has planar electrodes instead of bottom and top electrodes.

### 3.3. The Indirect Heating Structure of Phase-Change Switches

Hinnawy et al. from Carnegie Mellon University introduced the concept of a four-terminal indirectly heated phase-change RF switch utilizing GeTe thin films in 2013 [76]. This groundbreaking concept exhibited remarkable RF capabilities and stability, subsequently gaining significant attention and adoption among subsequent researchers. Termed as the first generation, this four-terminal RF phase-change switch demonstrated a 0.08 Ω·mm R_ON_ and a cut-off frequency (F_co_) of 1.0 THz. Notably, it was the pioneering application of an electrically isolated heater to successfully activate both states of an inline phase-change switch. With excellent RF performance and stability, this indirectly heated structure has been heavily applied by subsequent scholars. 

The process of the first generation is shown in Figure 8a, involving multiple steps: initially, SiO_2_ is formed on an HR-Si substrate, achieved through thermal growth or Plasma Enhanced Chemical Vapor Deposition (PECVD). Following this, NiCrSi thin film resistors are patterned using a stripping process. Subsequently, a dielectric barrier of Si_3_N_4_ is established using PECVD, with the contact window in this layer etched via dry Reactive Ion Etch (RIE). GeTe film is deposited at room temperature via RF sputtering and patterned using the lift-off method, leading to its crystallization. Later, the Ti/Au contact and interconnect metallization layers are patterned by the lift-off technique. Finally, an additional PECVD Si_3_N_4_ dielectric passivation is deposited, featuring dry-etched openings for electrical probing of the device. This design closely resembles a bottom-gated Field-Effect Transistor (FET), in which ohmic contacts are established with a channel material (the phase-change material or PCM) [77,78]. A thermally-isolated microheater is positioned beneath the chalcogenide channel, ensuring thermal coupling while maintaining dielectric isolation.

While the first generation (Gen1) process exhibited commendable RF performance, several critical issues persisted. Firstly, the NiCrSi heater, operating near its melting point (approximately 1200 K (approximately 1200 K), posed concerns regarding potential degradation of device performance and the likelihood of premature failure or damage [79]. Secondly, NiCrSi displays a negative resistance temperature coefficient, thereby diminishing the effective thermal width of the heater. Consequently, the electrical width of the melting region is reduced, subsequently curtailing the amorphization region and slowing down the switching speed. Lastly, amplifying the pulse power tends to decrease the lifespan of heater, directly impacting the device’s reliability. To address these issues, the second generation (Gen2) of phase-change switches emerged, as depicted in Figure 8b. In this upgraded process, tungsten (W) serves as the microheater material due to its notably high melting point (approximately ~3700 K) and its linear, positive resistance temperature coefficient [80]. In contrast to the first generation, the performance of the second-generation phase-change switch has witnessed significant enhancements.

**Figure 8 micromachines-15-00380-f008:**
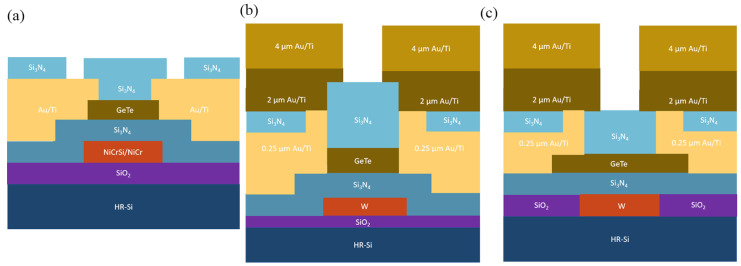
(**a**) The schematic cross-section of the Gen1 fabrication (adapted from [76]). (**b**) Schematic cross section of the Gen2 fabrication process (adapted from [77]). (**c**) Schematic cross section of the Gen3 fabrication process (adapted from [81]).

However, evident cracks along the microheater edges pose a significant challenge. This issue primarily arises due to stress induced by the deposition of GeTe films at room temperature followed by high-temperature annealing, leading to structural fissures. These cracks considerably impact the reliability of PCSs. To enhance RF performance while ensuring improved reliability, the third-generation process (Gen3) was developed, featuring a planar microheater architecture. Ti/TiN/W microheaters were crafted through an oxidation bath within an oxide trench using the damascene process. To maintain surface flatness, a chemical mechanical polishing technique (CMP) was employed on the top surface of the heater, as illustrated in Figure 8c. The thicker and flatter heater design of the third generation further enhances isolation between the control terminal and RF terminal. Moreover, it significantly reduces excitation pulse voltage requirements and lowers power consumption [81].

Although both types of switches have been investigated, indirectly heated PCSs are almost always used in current research on RF system applications, considering signal crosstalk and process integration issues. Table 2 lists some of the reported applications using indirectly heated PCSs.

#### 3.3.1. Substrate and Insulator Layer

For substrates suitable for phase-change switching, good heat dissipation, good high-frequency transmission characteristics, and good matching to the film are required. Typically for semiconductor substrates, an additional insulating layer is required to minimize charge crosstalk at high frequencies. In the current preparation of switching devices, the substrates that have been used are HR-Si, Sapphire, SiC, Alumina, etc. [19,28,32,79,95]. and insulators often used are SiO_2_, SiN_x_, AlN [23,28,75,96,97,98,99,100]. AlN can also be a heat spreader to reduce the consumption of PCSs [101]. Table 3 lists the commonly used combinations of substrate and insulator. In RF switches, the relative permittivity of substrate significantly influences overall RF performance. Concurrently, the thermal conductivity of the substrate plays a crucial role in determining the required drive voltage for the switch. Carriers within the semiconductor exhibit increased mobility, accelerating the response speed to the current, and consequently elevating conductivity levels. Therefore, an insulating layer is frequently employed to mitigate electrical crosstalk originating from the substrate at elevated frequencies.

#### 3.3.2. Heater Layer

In PCS applications, the phase-change material undergoes a transformation between two distinct states: the amorphous state characterized by high resistance (OFF) and the crystalline state characterized by low resistance (ON). To transition the phase-change material (GeTe) into the “OFF” amorphous state requires heating it above its melting temperature (730 °C) and subsequently rapidly cooling it (typically achieved with a thermal RC time constant less than 100 ns) to drop it below the crystallization temperature. The process of transitioning the same material into the “ON” state is comparatively less demanding, as heating it to a temperature below the melting point combined with a duration longer than 30 ns is often adequate to crystallize the material. Thus, taking the PC material into the amorphous state requires greater electrical power to achieve higher temperatures as well as switch design that permits cooling in a time constant shorter than 100 ns to prevent crystallization. This process places the greater constraints on the electrical drive as well as the switch architecture. A heater physically separated from the switch can be used to turn the switch on or off by producing a desired temperature. The heater must be capable of being driven, with sufficient power at high speed so that one can reach temperatures above 730 °C and cool sufficiently fast. The transient temperature of the heater is an issue of great concern to scholars, and finite-element modeler (FEM) simulation softwares like Comsol 5.5 and Ansys workbench 2019 are advantageous tools for these investigations. As can be seen from Figure 9, a microheater is required to generate heat by the joule effect and pass the heat to the PCM to implement the phase-change process.

Experiments have demonstrated that TiW, W, TiN, NiCrSi, NiCr, and Mo are all candidates for microheaters in PCSs, as shown Figure 10a. To achieve the melting point of GeTe, varying widths of heaters with different metal materials were simulated using Comsol 5.5 [26]. From the simulation result, a wider heater width generates a lower max temperature at constant thickness, and Mo generates more heat than W and Cr. Figure 10b illustrate the impact of varying heater widths on both MPA (minimum power to amorphize) and MPC (minimum power to crystallize), alongside the influence on the off-state capacitance (C_OFF_) [32]. The MPA escalates by 0.5 W for every micron increase in heater width, while the MPC rises by 0.1 W per micrometer. This increase in power can be attributed to the amplified heating volume, subsequently leading to an upsurge in thermal capacitance and thermal time constant. Consequently, to liquefy PC material within the same 100 ns voltage pulse, an elevation in the power supplied to the heater becomes necessary, corresponding the result of Figure 10a. Moreover, C_OFF_ experiences an increase of 4.3 fF per micrometer of increase of heater width. Therefore, a wider heater width increases the power consumption of PCSs. Figure 10c delineates the impact of the switch heater size on insertion loss (IL) and isolation (Iso). These measurement results conform to the trends observed in simulation outcomes from another report, as depicted in Figure 10d [26]. Therefore, a preliminary conclusion is drawn that a smaller heater width results in lower IL, and higher Iso and return loss (RL).

To address the necessity for efficient thermal conduction in phase-change RF switches, the resistive material constituting the heating layer assumes a pivotal role in the thermally induced phase-change process. Table 4 presents a compilation of candidate materials suitable for the heating layer of phase-change switches. It is noteworthy that, given compatibility with the process, the choice of material for PCSs relies on possessing the suitable heat capacity and resistivity.

#### 3.3.3. Dielectric Layer

The dielectric layer in a PCS functions to isolate the heating path from the RF path while facilitating efficient heat transfer to the GeTe film. It necessitates excellent insulation properties alongside high thermal conductivity. Effective insulation prevents interference or crosstalk between the DC activating pulse and RF transmission signal. Simultaneously, high thermal conductivity enables rapid heat conduction from the heating layer to the phase-change material, facilitating the phase change within the nanosecond-level pulse duration. Commonly employed dielectric materials in PCSs encompass SiO_2_, SiN_x_, AlN, and others. Table 5 shows the dielectric material used in PCSs. High resistivity and melting point are essential properties for dielectric layer materials to ensure effective insulation and thermal stability. Furthermore, research indicates that AlN with its high thermal conductivity, plays a beneficial role in enhancing switch speed and reducing DC drive power consumption [94].

Figure 11a,b depict that the implementation of AlN film as the dielectric layer exhibits lower MPA and heater temperature compared to SiN_x_ for a fixed F_co_ (cutoff frequency), which represents the FOM (figure of merit) and is defined as F_co_ = 1/(2π × R_ON_C_OFF_), where R_ON_ is the on-state resistance and C_OFF_ is the off-state capacitance [94]. Reduced MPA holds particular significance in numerous RF systems where generating high DC voltages proves either inaccessible or economically burdensome. To achieve a higher F_co_, elevating the heater temperature beyond half of the melting point of W (3665 K) is necessary, which can potentially degrade the dielectric layer. It can be seen that at the same F_co_, the switch with AlN has a smaller MPA than a switch using SiN_x_, which can reduce the power consumption of the device. In addition, AlN is more heat-proof compared to SiN_x_, contributing to enhanced reliability for PCSs. Figure 11c shows that both switches exhibit on-state resistance of 2 Ω up to 15 GHz. Altering the AlN barrier thickness from 105 nm to 170 nm results in a reduction of the off-state capacitance from 15 fF to 10 fF [95]. Figure 11d delves into the impact of dielectric layer thickness on RF performance via simulation. It becomes apparent that a thicker dielectric layer leads to reduced IL, increased Iso, and diminished RL. This relationship implies a trade-off between IL and Iso [32].

#### 3.3.4. RF Electrode

The RF electrode serves to establish a connection with the phase-change material (PCM), contributing to the resistance variation of GeTe and facilitating the transmission of RF signals. The gap length (L_pcm_) between the two RF electrodes significantly impacts the resistance of the R_ON_. According to the relationship R = ρL_pcm_/S (where ρ represents resistivity), R_ON_ is directly proportional to the L_pcm_. Consequently, a smaller L_pcm_ results in a reduced R_ON_. However, concurrently considering C_OFF_, determined by the equation C = εS/L_pcm_ (where ε represents relative permittivity), C_OFF_ is inversely proportional to the L_pcm_. Therefore, in the design of L_pcm_, considerations for both R_ON_ and C_OFF_ are essential. Achieving smaller R_ON_ requires a smaller gap, whereas the optimization of C_OFF_ necessitates a larger gap length. Balancing these factors becomes crucial in the design phase of L_pcm_ in RF electrode configuration.

Figure 12a,b show the MPA and MPC, respectively, for switches with L_pcm_ from 400 nm to 800 nm. As the L_pcm_ decreases, the MPA increases. This is because Au contacts are closer together and absorb more heat from the PC material. To compensate for the increased heat dissipation, more power is needed to raise the temperature of the PC material above its melting temperature [103]. Figure 12c,d show the R_ON_ and C_OFF_, respectively, extracted from 45 MHz to 20 GHz [32,103]. As expected, the R_ON_ decreases as the L_pcm_ decreases. As the L_pcm_ decreases from 800 nm to 400 nm, the R_ON_ decreases from 3 to 1.4 Ω. However, as R_ON_ decreases, C_OFF_ also increases from 11.7 to 15.4 fF. Figure 12d exhibits the simulation of RF performance for different gap lengths, and it can be seen that the smaller gap results in smaller insertion loss, but also smaller isolation and smaller return loss. There is a trade-off between R_ON_ and C_OFF_ as well as insertion loss and isolation.

Device architectures often require GeTe to be in contact with metal electrodes. Hence, the performance of these devices is notably susceptible to the influence of metal/GeTe interfaces. This scenario holds particular significance in GeTe-based RF switches, where studies indicate that contact resistance (R_c_) contributes significantly, accounting for 20–50% of the on-state resistance of the switch [105]. Table 6 demonstrates the test data on the contact resistance as well as the specific contact resistance of commonly used electrodes to GeTe. Considering the aforementioned data, achieving an ohmic contact between the electrode layer and GeTe becomes a critical inclusion in the optimization process of the switch design.

## 4. Conclusions

The utilization of chalcogenides in communication, storage, thermoelectricity, and various other domains is increasingly pervasive. Compared to representative RF switches, PCS boasts a host of advantages, including superior RF performance, high linearity, enhanced integration capabilities, non-volatile characteristics, and rapid switching speeds. Focused on phase-change RF switches, Ge_50_Te_50_ is selected as the core phase-change material through the screening of different stoichiometric composition. Based on the current researches, possible structures of GeTe in crystalline and amorphous states are detailed below.

The crystalline structure of GeTe film includes a distorted rock salt (R-GeTe) structure and a face-centered cubic (C-GeTe) structure. GeTe films deposited below 130 °C typically exhibit an amorphous state. Deposited GeTe films adopt a polycrystalline state within the temperature range of 130 °C to 250 °C. If grown on a NaCl or mica seed layer at nearly 250 °C, GeTe forms a single-crystalline with a rhombohedral structure. Above 250 °C, GeTe film presents a single-crystalline with an FCC structure, while the amorphous structure of GeTe film is surrounded by the combination of tetrahedral and defect octahedral configurations. Amorphous Ge_2_Sb_2_Te_5_ is characterized by even-folded ring structures, whereas amorphous GeTe has both even- and odd-numbered rings. The possible patterns of tetrahedral structure are as follows: 6.25% GeTe_4_, 25% GeTe_3_Ge, 37.5% GeTe_2_Ge, 2.25% GeTeGe_3_, and 6.25% GeGe_4_.

This paper offers a comprehensive comparative analysis of two prevalent structures of phase-change RF switches: direct heating structures and indirect heating structures. Although both types of switches have been investigated, indirectly heated switches are dominant in current research on RF system applications, considering signal crosstalk and process integration issues. Based on existing literatures, this paper also summarized material selection for each layer and the dimensions or thicknesses of switches within these layers. The resulting conclusions of RFPCSs are as follows:(1)The relative permittivity and of substrate significantly influences overall RF performance. Concurrently, the thermal conductivity of the substrate plays a crucial role in determining the required drive voltage for the switch.(2)The heater must be capable of being driven, with sufficient power at high speed so that one can reach temperatures above 730 °C and cool sufficiently fast. A wider heater width would increase the power consumption of the PCS. A preliminary conclusion is drawn that a smaller heater width results in lower insertion loss (IL), and higher isolation (Iso) and return loss (RL).(3)The thicker dielectric layer could decrease C_OFF_ but increase power consumption for a higher MPA. AlN could lower power consumption for PCSs compared to a device using SiN_x_. It becomes apparent that a thicker dielectric layer leads to reduced IL, increased Iso, and diminished RL. This relationship implies a trade-off between IL and Iso.(4)There is a trade-off between R_ON_ and C_OFF_ and insertion loss and isolation for the gap width of the RF transmission electrodes. Obviously, as the gap decreases, the insertion loss decreases, but so does the isolation and return loss. In addition, the effect of contact resistance due to the interaction of the electrodes with the GeTe must be considered.

## 5. Outlook

Over the past decade, RFPCS technology has rapidly developed for application in reconfigurable RF front-end systems. The excellent features of easy integration and low power consumption allow PCSs to make a big impact up to millimeter-wave frequencies. There has been a succession of excellent works on RF components based on phase-change switching application. Although the current GeTe-based PCSs have many excellent features, there are still a lot of areas that can be improved, such as switching speed, power handling capability, endurance, etc. In this review, the influence and role of each layer on switching performance has been analyzed, and it is believed that this review can provide a corresponding reference for enhancing performance in future communications.

After extensive research efforts, significant strides have been achieved in advancing the development and utilization of GeTe RFPCSs. Progress has been notable across structural design, materials utilization, and process optimization, transitioning from the initial direct heating configuration to the more refined indirect heating architecture. This latter approach has undergone iterative changes from its initial introduction to its current state as a third-generation model. With the development of integration process and micron/nano scale fabrication technology, it will certainly be more perfect. The future process direction of RFPCSs is mainly along the monolithic or heterogeneous integration. The GeTe phase-change RF switches exhibit three pivotal characteristics: high cutoff frequency, non-volatility, and compatibility with back-end processes. These distinctive features render these switches highly advantageous tools for augmenting flexibility across a spectrum of technological domains, including millimeter-wave networks, sub-6G wireless systems, reconfigurable architectures, and passive circuits. Moreover, these switches serve as integral components in the construction of reconfigurable devices, employed as discrete packaging elements. They are anticipated to supplant commonly utilized PIN diodes in specific scenarios, representing a rare occurrence in their application as discrete packaging components within reconfigurable device architecture. In military applications, leveraging GeTe phase-change RF switches facilitates the development of customized field-programmable gate arrays (FPGAs) designed specifically for constructing adaptable devices. These advancements hold the potential for cost savings within military communication, electronic warfare, and intelligence systems, as they capitalize on the capabilities offered by field-programmable gate array (FPGA) RF front-end components grounded on GeTe phase-change RF switches.

## Figures and Tables

**Figure 2 micromachines-15-00380-f002:**
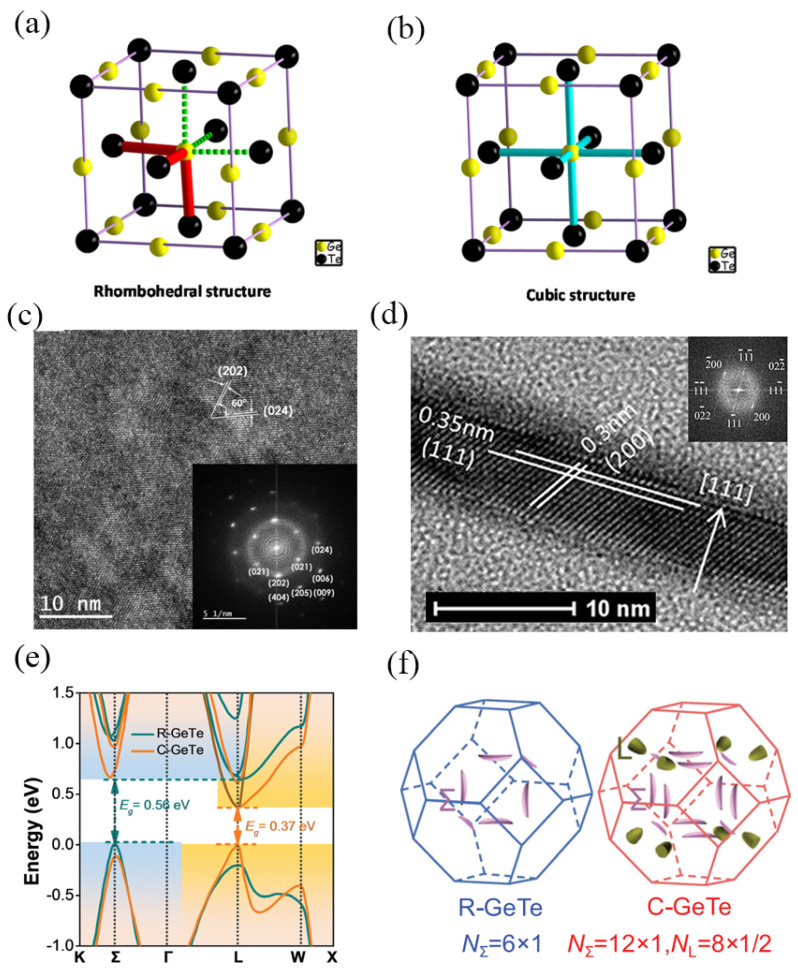
Structure of crystalline GeTe: (**a**) rhombohedral configuration, (**b**) cubic configuration (adapted from [48]). (**c**) TEM of GeTe film deposited and annealed at 573 K for 24 h (inset: corresponding FFT) (adapted from [49]). (**d**) High-resolution TEM images crystalline GeTe film annealed at 673 K (inset: corresponding FFT) (adapted from [50]). (**e**) Calculated band structures of R-GeTe and C-GeTe, respectively (adapted from [51]). (**f**) Crystal structure of α-GeTe in hexagonal setting (note that the space-group symmetry is not hexagonal due to the presence of a threefold, rather than sixfold, rotation axis) (adapted from [52]).

**Figure 3 micromachines-15-00380-f003:**
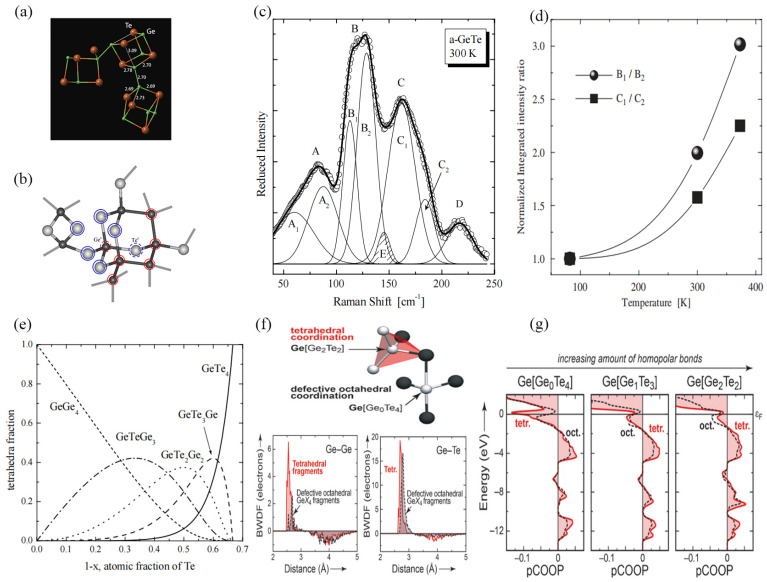
(**a**) The panel shows Ge and Te atoms that are involved in tetrahedral Ge configurations using DFT simulations (adapted from [56]). (**b**) Possible mechanism for the amorphous-to-crystal transition (adapted from [57]). (**c**) Quantitative description of the vibrational modes with the aid of a Gaussian line fit procedure (adapted from [57]). (**d**) Normalized intensity ratio of the B_1_, B_2_ and C_1_, C_2_ bands as a function of temperature designating the increase of Te-rich tetrahedra with respect to the Ge-rich tetrahedra as temperature is raised (adapted from [59]). (**e**) Relative distribution of the various types of GeTe_n_G_1−n_ tetrahedra, as a function of the chalcogenides’ atomic fraction. The smaller red dots represent Ge ions and the larger yellow dots represent Te ions (adapted from [59]). (**f**) Upper part: Structural fragment drawn to emphasize the different coordination motifs found in the amorphous phase. Bottom parts: BWDFs of homopolar (Ge-Ge) and heteropolar (Ge-Te) contacts, according to the coordination environments (adapted from [60]). (**g**) P-COOP analysis resolved according to GeX4 motifs in one of the structural snapshots. P-COOPs (per bond) are averaged separately for tetrahedral and defective octahedral motifs (adapted from [60]).

**Figure 4 micromachines-15-00380-f004:**
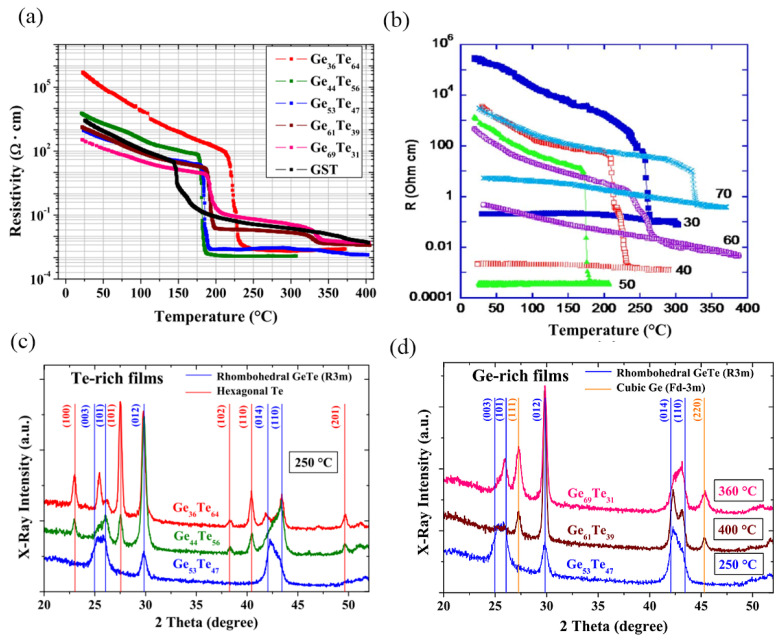
(**a**) Temperature-dependent resistivity measurements performed on 100 nm thin Ge_x_Te_100−x_ films. The results reported are for a heating rate of 10 °C/min (adapted from [64]). (**b**) R–T curve for GeTe films of variable composition, with 30, 40, 50, 60, 70 at. % Ge (adapted from [65]). XRD patterns of (**c**) Te-rich samples and (**d**) Ge-rich samples (adapted from [64]). The smaller red dots represent Ge atoms and the larger yellow dots represent Te atoms.

**Figure 5 micromachines-15-00380-f005:**
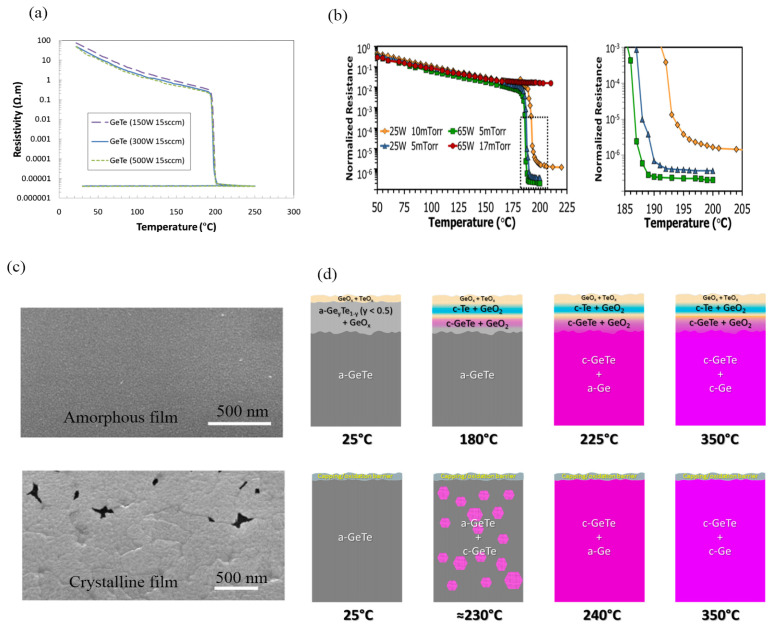
(**a**) Sheet resistance change in GeTe films annealed at elevated temperature with different sputtering power and Ar flow (adapted from [70]). (**b**) Normalized resistance in GeTe films at elevated temperature with different sputtering power and deposition pressure, and enlarged image of dashed areato show the transition temperature and final resistance value. (adapted from [71]). (**c**) SEM of amorphous and crystalline GeTe film (adapted from [71]). (**d**) Schematic representation of the surface-oxidized GeTe film during annealing with elevated temperatures and for the capped GeTe film (adapted from [72]).

**Figure 6 micromachines-15-00380-f006:**
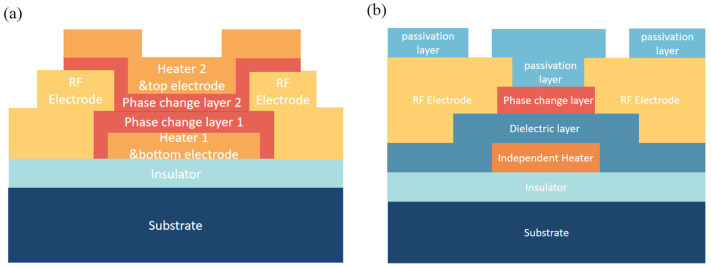
(**a**) Typical directly heated switch model illustrated (**b**) Typical indirectly heated switch model.

**Figure 7 micromachines-15-00380-f007:**
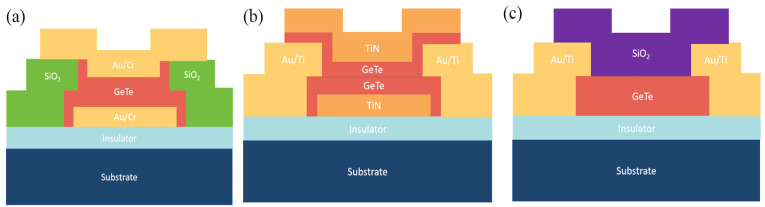
(**a**) Two-port directly heated model (adapted from [22]). (**b**) Four-port directly heated model [23]. (**c**) The image of the four-port planar structure GeTe direct heating switch (adapted from [26]).

**Figure 9 micromachines-15-00380-f009:**
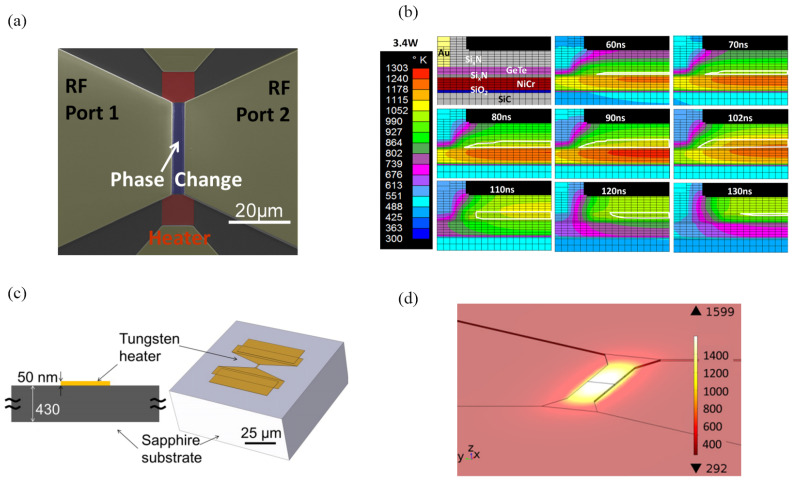
(**a**) Colored SEM of a GeTe phase-change RF switch (adapted from [79]). (**b**) Simulated progression of the heat pulse from the heater into the GeTe layer, with the melt zone delineated by the white borders by Ansys (adapted from [79]). (**c**) Cross-sectional view of the device and Comsol perspective view of the device (adapted from [102]). (**d**) Perspective view of the simulated temperature map of the heater (adapted from [102]).

**Figure 10 micromachines-15-00380-f010:**
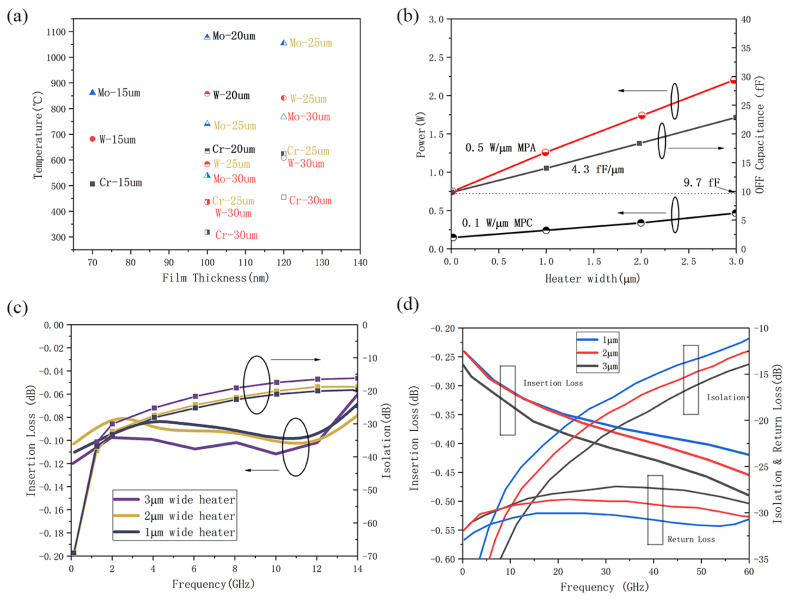
(**a**) Simulated temperatures for 15, 20, and 25 nm-wide chromium, tungsten, and molybdenum microheaters with thicknesses of 70, 100, and 150 nm (data from [32]). (**b**) MPA, MPC, and Extracted C_OFF_ for 20 μm switches with heater widths of 1, 2, and 3 μm (data from [103]). (**c**) Measured insertion loss and isolation with heater widths of 1, 2, and 3 μm from 45 MHz to 14 GHz (data from [103]). (**d**) Insertion loss, isolation and return loss EM simulation of the PCM switch with heater widths of 1, 2, and 3 μm (Data from [32]).

**Figure 11 micromachines-15-00380-f011:**
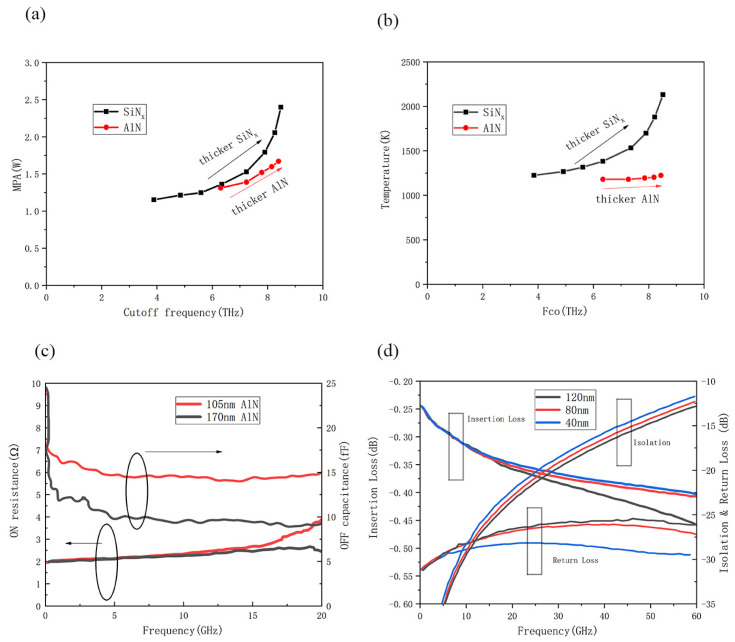
(**a**) Simulated MPA and (**b**) simulated maximum heater temperature as a function of F_CO_ for a 1.06 µm-wide heater, where SiN_x_ thickness ranged from 20 nm to 155 nm while AlN was between 70 nm and 200 nm (data from [104]). (**c**) R_ON_ and C_OFF_ of 105 nm and 170 nm AlN film from DC-20 GHz (data from [104]). (**d**) Insertion loss, isolation and return loss EM simulation of the PCM switch with a variation in barrier thickness (Data from [32]).

**Figure 12 micromachines-15-00380-f012:**
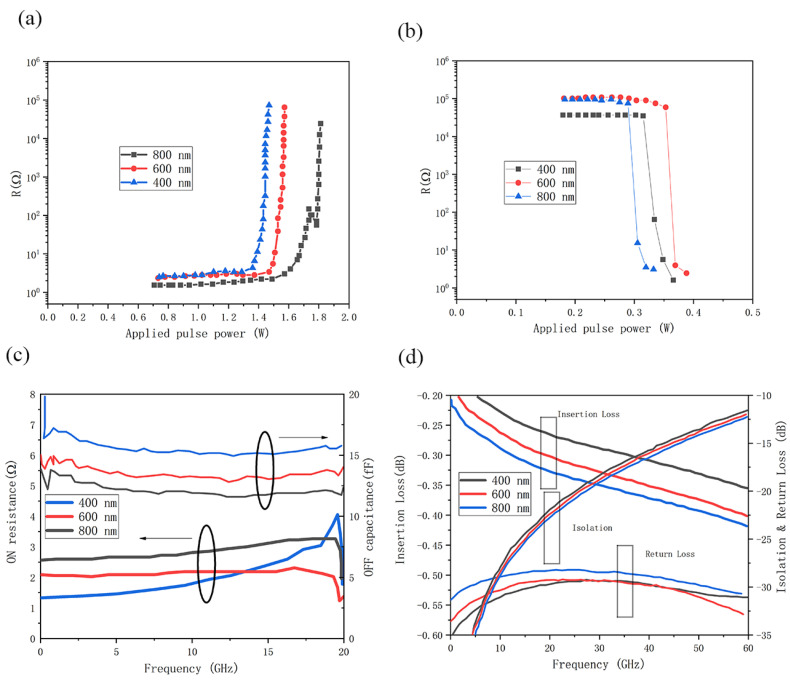
(**a**) MPA and (**b**) MPC results for 20 μm-wide switches with RF gaps ranging from 400 to 800 nm in length (data from [103]). (**c**) Extracted R_ON_ and C_OFF_ of switches with varying L_pcm_ from 45 MHz to 20 GHz (data from [103]). (**d**) Insertion loss, isolation, and return loss EM simulation of the PCM switch with a variation in channel length (data from [32]).

**Table 1 micromachines-15-00380-t001:** Relevant properties of Ge_2_Sb_2_Te_5_, GeTe, Sb_x_Te_y_, and VO_2_ for PCS.

Phase-Change Material	Ge_2_Sb_2_Te_5_	GeTe	Sb_x_Te_y_	VO_2_
Crystallization temperature T_c_ (°C)	140	190–210	145–150	65–80
Crystalline resistivity (Ω·cm)	10^−1^	10^−4^	10^−2^	10^−1^
R_on_ (Ω)	10	0.9	4.5	10
R_off_ (Ω)	150 k	35.3 k	35 k	10 k
Switching time	270 μs	2 μs	1.3 μs	0.4 μs
Non-volatile	Yes	Yes	Yes	No
Ref.	[41]	[16,17,18,19,20,21,22,23,24,25,26,27,28]	[39,40]	[42,43,44]

**Table 2 micromachines-15-00380-t002:** Excellent works on RF components based on GeTe film phase-change switches.

Components	Range (GHz)	RF Performance *: IL (Insertion Loss), ISO (Isolation), RL (Return Loss), PH (Power Handling), etc.
X-Band Reconfigurable bandpass filters [82]	7.45–8.07	IL: 2.6–3.2 dB IIP_3_ **: 30 dBm Tuning speed: <6 μs
6-bit latching switched capacitor bank [83]	2–7	Capacitance tuning range: 0.14 pF to 8 pF Capacitance ratio: 58:1 Tuning speed: 1.4 μs
4-bit Latching Variable Attenuator [84]	24–32	Maximum attenuation: 37 dB Minimum attenuation: 4.7 dB RL: >20 dB
Switched K-Band Tunable Reflective Load [85]	DC-60	The resonance: <18 GHz or > 26.5 GHz Phase shift: 30° at 18 GHz and up to 45° at 26.5 GHz
3-bit switched True-Time-Delay phase Wideband shifters [86]	26–34	Phase shifter at 30 GHz: 170° (ps-a), 173° (ps-b) IL_max_: 4.9 dB (ps-a), 4.7 dB (ps-b) RL_min_: 16.5 dB (ps-a), 14 dB (ps-b)
Integrated Wideband Digital Switched Attenuator [87]	26–34	Attenuation range: 24.1 dB IL: 3.9 dB, RL_min_: 13 dB PH: 35.5 dBm, IIP_3_: 4 1 dBm
Reflection-Type Phase Shifter With 8-Bit Switched Phase Tuning [88]	26–30	Phase shifter at 30 GHz: 280° IL_max_: 6 dB RL_min_: 14.2 dB;
Monolithic Band Reject Circuit [89]	1–8	Four states of this circuit: reject f_1_ at 2.6 GHz; reject f_2_ at 5.8 GHz; reject f_1_ and f_2_ at the same time; no rejection IL < 2.2 dB RL > 17 dB
Switch matrix [90]	DC-60	IL < 3 dB, RL > 14 dB, ISO > 20 dB
R-type and C-type Switch [91]	DC-30	C-type: IL < 0.75 dB, RL > 18 dB R-type: IL < 1.1 dB (state I/II), <1.5 dB (state III), RL > 18 dB, ISO > 25 dB
T-type switch [92]	DC-67	IL < 0.6 dB, RL > 20 dB, ISO > 20 dB
DPDT (Double-Pole, Double-Throw) Switch [93]	DC-20	IL < 2 dB (0–5 GHz), IL < 4 dB (@20 GHz) ISO > 17.5 dB (0–20 GHz)
SPNT (Single-Pole, N-Throw) Switches [94]	DC-40	SP2T: IL < 0.5 dB (0–20 GHz), IL < 0.7 dB (0–40 GHz) ISO > 26 dB (0–40 GHz) SP4T: IL < 0.6 dB (0–20 GHz), IL < 1 dB (0–40 GHz) ISO > 25 dB (0–40 GHz) SP9T: IL < 0.5 dB (0–20 GHz), IL < 1.5 dB (0–40 GHz) ISO > 28 dB (0–40 GHz)

* IL: Insertion Loss; ISO: isolation; RL: Return Loss; PH: Power Handling. ** IIP3: third-order intercept point.

**Table 3 micromachines-15-00380-t003:** Common substrates and insulating materials used in RF switches.

Substrate (Sub)	Thermal Conductivity (W/m·K)	Relative Permittivity of Sub	Insulator	Refs.
HR-Si	150	11.9	SiO_2_	[23,95,100]
HR-Si			SiN_x_	[99]
HR-Si			AlN	[19]
SiC	120–490	10	SiO_2_	[79]
Alumina	18–35	11.5	SiO_2_	[28]
Sapphire	25–40	9.8	N/A	[32]
GaAs	46	12.5	N/A	[97]
GaN	9	150	N/A	[98]

**Table 4 micromachines-15-00380-t004:** Materials used as microheater in GeTe-based PCSs.

Material	Thermal Conductivity (W/m·K)	Melting Point (°C)	Resistivity (Ω·m)	Type	Refs.
Mo	142	2610	5.2 × 10^−8^	indirect	[93]
AlCu	104	670	2.5 × 10^−8^	direct	[21,22]
W	174	3390	5.0 × 10^−8^	indirect	[26,32,84,91]
NiCr	12.2	1350	1.1 × 10^−6^	indirect	[28]
NiCrSi	-	-	-	indirect	[18,67,68,93]
TiN	19.2	2950	2.0 × 10^−7^	indirect	[27]
TiN	19.2	2950	2.0 × 10^−7^	direct	[19,20]

**Table 5 micromachines-15-00380-t005:** Dielectric materials used in PCSs.

Dielectric Material	Thermal Conductivity (W/m·K)	Melting Temperature (°C)	Resistivity (Ω·m)	Refs.
SiO_2_	1.4	1700	10^12^~10^14^	[17,18,19,20,21,22,23,31]
SiN_x_	3	1900	10^9^~10^12^	[90]
AlN	140	2500	10^10^~10^14^	[23,102,104]

**Table 6 micromachines-15-00380-t006:** Contact resistance and specific contact resistivity of GeTe and metal [105,106].

Contact Metal	Contact Resistance R_c_ (Ω·mm)	Specific Contact Resistivity ρ_c_ (Ω·cm^2^)	Sheet Resistance R_sh_ (Ω/sq)
Pd/Ti/Pt/Au	0.0036 ± 0.002	(3.7 ± 0.2) × 10^−9^	36
Mo/Ti/Pt/Au	0.0035 ± 0.001	(3.6 ± 0.7) × 10^−9^	38
Ni/Ti/Pt/Au	0.016 ± 0.003	(6.4 ± 2.2) × 10^−8^	40
Sn/Fe/Au	0.0037 ± 0.001	(5.2 ± 0.6) × 10^−9^	43
Ti/Pt/Au	0.034 ± 0.0028	(3.13 ± 0.6) × 10^−7^	38
TiW/Al	0.0054 ± ⋯	(1.43 ± ⋯) × 10^−8^	…
Cr/Pt/Au	0.0055 ± 0.002	(9.6 ± 0.8) × 10^−9^	36
W/Al	0.0054 ± ⋯	(1.4 ± ⋯) × 10^−8^	…

## Data Availability

No new data were created or analyzed in this study. Data sharing is not applicable to this article.

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
