# Peer review of "A Review on Material Selection Benchmarking in GeTe-Based RF Phase-Change Switches for Each Layer"

_micromachines, 2024, doi:10.3390/mi15030380_

Round 1

Reviewer 1 Report

Comments and Suggestions for Authors

The authors of the manuscript titled "A Review on Material Selection Benchmarking in GeTe-Based RF Phase Change Switches for Each Layer" have comprehensively examined the current research progress in RF switches based on GeTe. They have conducted a comparative analysis of the stack structure in both indirectly and directly heated phase-change RF switches, elucidating the functionalities of each layer. This review holds potential interest and utility for the RF device community.

While the scientific content remains unaddressed, several editing errors have been identified and should be rectified prior to publication:

-          The order of acronym definition and usage is inconsistent, with instances of using undefined acronyms or employing the full name instead of an already defined acronym.

-          Superscript/subscript symbols are inconsistently applied and may lead to confusion. Please verify the size and format for clarity.

-          It is imperative to confirm the acquisition of permission to use figures from references.

-          The overall figure quality is suboptimal; efforts to enhance it are encouraged.

-          The micro symbol is at times confusing, such as in line 343, Table 2.

-          Inconsistencies exist in the format for indicating the author of a cited paper in the text. Adopt a consistent approach, using either the last name or full name.

-          On page 3, line 86, the phrase "some interesting" is repeated.

-          On page 6, lines 179-182, the sentence "This structural arrangement undergoes a remarkable transformation throughout the crystallization process, transitioning from a tetrahedral form to a defect-free octahedral form." is duplicated.

-          On page 11, line 320, clarify the unit for "Ohm/??."

-          On page 13, section 3.2, in lines 389-390, only one reference is cited despite numerous tagged references. Additionally, the same reference is reiterated in line 396.

-          On page 14, line 435, correct the figure reference to Fig. 7 instead of Fig. 6.

-          On page 16, lines 468-472, rectify grammatical errors in these sentences.

-          On page 19, Table 3, use "N/A" instead of "NA."

Comments on the Quality of English Language

Refer to the general comments

Reviewer 2 Report

Comments and Suggestions for Authors

This manuscript reviews the characteristics of GeTe films applied to RF phase-change switches (RFPCS) based on direct and indirect heating structures. It focuses on the prototypical structure used in indirectly heated RFPCS as a reference and analyzes the intrinsic properties of each material layer and the rationale behind material selection. The design size of each material layer of the device and the RF performance are summarized. Overall, the paper presents a comprehensive comparative analysis of the two prevalent RFPCS based on direct and indirect heating structures and summarizes the roles of the building blocks therein. At last, an outlook is given for future applications.

This review supplies a wealth of information the different building blocks in GeTe based radio frequency phase change switches (RFPCS). It compares the properties and suitability of the individual components (beside the phase change material itself), such as the substrate, the insulator layer, the heater layer, the dielectric layer and the RF electrode for the RFPCS application and also with respect to the different direct and indirect heating structures. The review is very valuable. It certainly should be published in “Micromachines” after a few comments and remarks have been addressed.

It is strongly recommended, that a native speaker corrects the language of the manuscript rigorously. Furthermore, there are a number of typos (see for example page 11, line 341 (“gallium” instead of “germanium” telluride) and some repetitions (see for example page 3, line 86; page 6, lines 179-183) and ). The manuscript is certainly worth the effort.

Introduction: Since the manuscript is a review, it should give a short overview on different types of phase change switches citing other groups dealing with them:

(1) Stegmaier, M.; Ríos, C.; Bhaskaran, H.; Wright, C. D.; Pernice, W. H. P. Nonvolatile All-Optical 1 × 2 Switch for Chipscale Photonic Networks. Adv. Opt. Mater. 2017, 5 (1), 1600346. https://doi.org/10.1002/adom.201600346.

(2) Tanaka, D.; Shoji, Y.; Kuwahara, M.; Wang, X.; Kintaka, K.; Kawashima, H.; Toyosaki, T.; Ikuma, Y.; Tsuda, H. Ultra-Small, Self-Holding, Optical Gate Switch Using Ge_2Sb_2Te_5 with a Multi-Mode Si Waveguide. Opt. Express 2012, 20 (9), 10283. https://doi.org/10.1364/OE.20.010283.

(3) Zhang, Y.; Chou, J. B.; Li, J.; Li, H.; Du, Q.; Yadav, A.; Zhou, S.; Shalaginov, M. Y.; Fang, Z.; Zhong, H.; Roberts, C.; Robinson, P.; Bohlin, B.; Ríos, C.; Lin, H.; Kang, M.; Gu, T.; Warner, J.; Liberman, V.; Richardson, K.; Hu, J. Broadband Transparent Optical Phase Change Materials for High-Performance Nonvolatile Photonics. Nat. Commun. 2019, 10 (1), 4279. https://doi.org/10.1038/s41467-019-12196-4.

(4) Mikulics, M.; Hardtdegen, H. H. Fully Photon Operated Transmistor / All-Optical Switch Based on a Layered Ge1Sb2Te4 Phase Change Medium. FlatChem 2020, 23, 100186. https://doi.org/10.1016/j.flatc.2020.100186.

(5) Zheng, J.; Zhu, S.; Xu, P.; Dunham, S.; Majumdar, A. Modeling Electrical Switching of Nonvolatile Phase-Change Integrated Nanophotonic Structures with Graphene Heaters. ACS Appl. Mater. Interfaces 2020, 12 (19), 21827–21836. https://doi.org/10.1021/acsami.0c02333.

(6) Ghosh, R. R.; Dhawan, A. Integrated Non-Volatile Plasmonic Switches Based on Phase-Change-Materials and Their Application to Plasmonic Logic Circuits. Sci. Rep. 2021, 11 (1), 18811. https://doi.org/10.1038/s41598-021-98418-6.

Comments on the Quality of English Language

It is strongly recommended, that a native speaker corrects the language of the manuscript rigorously.
